# Eliminating Measles: Factors That Contribute to Re-Establishing Transmission

**DOI:** 10.3390/vaccines13111125

**Published:** 2025-10-31

**Authors:** Emily Gibson, David N. Durrheim, Patrick O’Connor

**Affiliations:** 1Hunter New England Public Health, Wallsend, NSW 2287, Australia; 2School of Medicine and Public Health, University of Newcastle, Callaghan, NSW 2308, Australia; 3Independent Researcher, New York, NY 11225, USA; oconnorpatrickm@gmail.com

**Keywords:** measles, surveillance, vaccination coverage, elimination, eradication, epidemiology

## Abstract

**Background/Objectives**: Since 2016, 14 countries previously verified to have eliminated endemic measles transmission have lost their elimination status. To explore whether there were identifiable risk factors for re-establishing measles transmission in these countries, a narrative review of the literature was undertaken. This identified several potential factors: national vaccination coverage, age at first and second measles-containing vaccine dose, the country’s socioeconomic status, surveillance quality, bordering countries endemicity status, supplementary vaccination campaigns just prior to elimination verification, population density, island status, and tourism volume. **Methods**: World Health Organization (WHO) vaccination and surveillance data were utilized, together with public-facing datasets from the World Bank and the United Nations. An exploratory analysis was undertaken with descriptive epidemiology used for comparing countries where elimination was sustained with those where endemic measles transmission was re-established. Regression modelling was then undertaken with those factors identified as of potential importance. **Conclusions**: Both descriptive epidemiology and regression modelling indicated that the most important factor is vaccination coverage, although the quality of vaccination coverage estimates from different data sources should be considered. Low–middle income socioeconomic status and bordering endemic countries increased the risk of re-establishment of measles transmission for verified countries. Without coordinated global efforts towards measles eradication, it will be challenging for some countries to maintain elimination.

## 1. Introduction

Measles is one of the most infectious diseases, with an estimated reproductive number of 12 to 18 [1]. Eradication is feasible, as humans are the only known reservoir, infection from disease confers life-long immunity, and an effective intervention (vaccine) exists [2]. However high two-dose vaccination coverage is required to achieve elimination [3]. Outbreaks in low measles-containing vaccine (MCV) coverage settings can be difficult to control, given the highly infectious nature of the virus.

Achieving elimination verification status is based on three main criteria: documentation of the interruption of measles transmission for at least 36 months, the presence of a high-quality surveillance system that is both sensitive and specific, and genotyping evidence supporting the interruption of endemic transmission [4]. Countries that have achieved elimination are required to maintain high MCV coverage and maintain sensitive surveillance to ensure their ability to rapidly detect and respond to outbreaks. The measles discard rate (the number of rash and fever cases per 100,000 population that are tested for measles and then discarded due to a negative result) is used as a proxy for surveillance quality [4].

Since 2016, of 205 WHO countries and territories, 93 had been verified to have interrupted measles transmission for at least 36 months. Of those 93 countries and territories (see note), 14 have experienced measles transmission being re-established for greater than 12 months and have thus lost their elimination status. Of the 14, 4 subsequently regained their elimination status, but that is not considered in this analysis as our focus is on primary re-establishment of transmission. Therefore, 79 countries make up the never lost verification (NLV) category, and 14 are considered as re-established (Figure 1).

We used available data to explore the associations with losing verification status. If modifiable factors could be identified, then countries could focus on these to ensure that elimination could be maintained. Similarly, identification of non-modifiable factors that placed countries at risk could help to inform broader policy decisions.

Global and regional measles resurgences test the elimination status of countries. Of note, a cluster of countries lost their elimination status during the 2018/2019 global resurgence with a particular impact in Europe. This outbreak saw large increases in measles incidence for Romania, France, Italy, Greece, and the United Kingdom, with these countries accounting for over 86% of notified cases in the European Union in 2018. Romania had been experiencing an outbreak since 2016 that was ongoing in 2018 and may have been driving transmission elsewhere through exporting the virus in travellers [5].

Large outbreaks in 2018/2019 and 2024/2025 in the Philippines and Viet Nam, respectively, seeded large numbers of cases globally and tested the robustness of eliminated countries’ statuses.

The authors hypothesized that factors related to vaccination, such as age at first and second dose, level of reported routine population vaccination coverage, and vaccine brand, might be important risk factors for the re-establishment of transmission. Other factors considered, based on a narrative review of the available literature, included population density [1,6,7,8,9], tourist volume [10,11,12], and whether countries bordered endemic countries [13,14,15,16] or were island nations [17,18,19,20]. A country’s socioeconomic status and measles mass vaccination campaigns in the period just prior to verification were also considered important factors to consider.

## 2. Materials and Methods

### 2.1. Data Sources

Country-level data from the World Health Organization and other publicly available datasets were collated to provide a dataset for analysis spanning 2012–2023 (verification commissions commenced formal review in 2012). A complete list of the datasets used is shown in Table 1.

WUENIC coverage estimates were selected for use in the analysis, rather than a country’s administrative reports. WUENIC vaccination coverage is estimated for each country for first and second routine measles-containing vaccine doses (annual WHO and UNICEF estimates of national immunization coverage WUENIC dataset). These data have been produced since 1999, and for the purposes of this analysis, the coverage estimate for the year of re-establishment was used (WHO/UNICEF estimates of national immunization coverage (https://www.who.int/news-room/questions-and-answers/item/who-unicef-estimates-of-national-immunization-coverage#:~:text=The%20WUENIC%20data%20provides%20information,or%20under%2Dimmunized%20communities%20persist (accessed on 29 October 2025))). There are limitations to the WUENIC data, namely timeliness, reliability, and completeness of the source data (impacting accuracy), and bias affecting surveys, such as priority communities not being counted thoroughly [21]. However, it was selected over routinely reported country administrative coverage for use in the analysis as it is recognized as the most reliable comparable coverage data available [22].

Data were merged and cleaned and descriptive epidemiological analysis was conducted in R version 4.4.1 for all countries that had been formally verified to have eliminated endemic measles transmission by their Regional Verification Commission. A single timepoint was selected for all countries. For re-established countries, this was the year of re-establishment in most instances. For NLV countries, this was 2022 in most instances. Exceptions were made where the logical choice was the midpoint in the data (for example, population numbers and annual tourism numbers are 2018 estimates for NLV countries). The working dataset for the analysis is provided in the Appendix A.

### 2.2. Study Design

Statistical analysis was performed in two parts, by modelling the results of the risk of re-establishment associated with 10% decreases in MCV coverage for dose one and dose two, considered separately. The results are reported as a risk ratio with a 95% confidence interval.

Logistic regression was then performed using the GLM command from base R and plotted using the GGPlot2 package. Four models were run in total, two multivariate and two univariate. Models one and two used derived first and second dose coverage categories, <85%, 85–89%, 90–94%, and ≥95%. Model one was multivariate and included all six factors considered important for the model (see Section 2.3). An assessment of correlation was then performed using the corrplot package. Unsurprisingly, collinearity was seen across the country and the derived coverage category variable, which used the WUENIC estimation. Therefore, in model two, the derived variable (WUENIC estimate) was only used. Models three and four explored the findings of model two; that first dose coverage was negatively correlated with re-establishment. For model three, a simplified vaccination coverage variable was derived, which used two groups, >90% and ≤90%. This was used to ensure that the previous coverage category variable (with four steps) was not too granular for the small dataset under consideration. As the results for this model were similar to model two, a fourth, experimental model considered the authors’ hypothesis that data quality might be contributing to counterintuitive results.

### 2.3. Model Inputs

The results of the initial descriptive epidemiological analysis indicated that eight factors were important (Table 2). These factors were used in the generalized linear models alongside derived vaccination coverage variables, described above.

## 3. Results

Fourteen countries verified as having achieved interruption of endemic measles transmission experienced re-establishment of endemic measles transmission between 2016 and 2023 (Figure 2). Nine of these countries experienced re-establishment between 2018 and 2019, and eight of these were in the European Region. Four of these countries were subsequently reverified to have interrupted transmission: Czechia, Greece, the United Kingdom, and Venezuela.

### 3.1. Socioeconomic Status

Socioeconomic status was explored as a factor in the re-establishment of measles transmission. Six of the fourteen countries (43%) were categorized as high income, compared to 55% (n = 44) of countries that maintained their verification status being high income. Four (29%) re-established countries were classified as lower-middle income countries (LMIC) as compared to seven (9%) of the NLV countries. A comparatively greater proportion of the re-established countries were designated lower-middle income (Figure 3).

### 3.2. Bordering a Measles Endemic Country

A statistically significant association was found between bordering a country endemic for measles and re-establishment of measles transmission (*p*-value 0.005, Figure 4). Island status was also considered; 14% (n = 2) of re-established countries were islands, compared to 29% (n = 23) of NLV countries.

### 3.3. Population Density

Macao SAR was the mostly densely populated territory (19,967 people per square kilometre, Figure 5). Two re-established countries, Sri Lanka and the United Kingdom, were included in the 25 most densely populated countries, with 330 people per square kilometre and 273 people per square kilometre, respectively. Table 3 indicates that the median population density in the re-established countries is comparatively lower than in the NLV countries.

### 3.4. The Role of Tourism

Where measles elimination has been verified, the source of the virus is obviously imported cases. Each tourist visit from a country where measles is circulating represents a potential viral importation. Although ideally data on country of origin and other countries visited should be analyzed, these are not readily available. As a proxy for tourism risk of introduction, a tourist-to-resident ratio was derived from raw tourism numbers for each country. The tourist-to-resident ratio is a measure of tourist “pressure”. A ratio above one indicates more tourists than residents per annum. Five (28%) of countries in the re-established dataset have a ratio above one (see Figure 6), compared to 39 (49%) of countries in the NLV group. No significant association was found between tourist pressure and re-establishment (Fisher’s exact *p*-value = 0.5). Raw tourism numbers were therefore utilized in the analysis in place of the tourist-to-resident ratio.

### 3.5. Vaccination Timing and Coverage

Figure 6 indicates a reasonable consistency in the age at which the first measles-containing vaccine (MCV) dose is scheduled, but inconsistency in the age at which the second MCV dose is scheduled. The timing of the second dose is considered important for seroconversion and limiting waning immunity and is influenced by administrative considerations, such as the number of health care visits required to receive childhood immunizations. The actual age that children are vaccinated is variable but is not available for most countries [23]. Age at second dose is comparably variable in countries that had not lost their elimination status and those that had (see Figure 7).

The quality of surveillance data is usually assessed by using the measles-like illness (non-measles fever and rash) discard rate. The minimum discard rate recommended is >2/100,000, or 2 non-measles fever/rash cases investigated and discarded per 100,000 population [4] (WHO, 2013). Nine of the re-established countries met the minimum discard rate, with four having a higher discard rate (>2). Albania had a discard rate of 44 per 100,000, which was the second highest value of all NLV and re-established countries, with Bahrain the highest at 76 (NLV). Of note, Albania was experiencing a large outbreak with high rates of measles notifications, confirmed and discarded, that likely contributed to the high discard rate. Figure 8 describes this relationship, noting that zero values are excluded from the graph (n = 19, 2 re-established: 17 NLV), and fifteen countries (2 re-established: 13 NLV) did not have a discard rate available for assessment.

Reasonable agreement between first and second dose coverage for NLV and re-established countries exists, with some outliers (see Appendix B).

Reasonably good agreement between the two groups of countries in MCV coverage is demonstrated, and so further analysis was undertaken to better understand the risk of lower MCV coverage (Table 4). A risk ratio was calculated for 10% decreases in coverage of first and second dose, for the <90%, <80%, and <70% categories. Although an even more granular analysis (e.g., 5%) would have been preferable, this was not possible due to the relatively small number of re-established countries. The <70% first dose coverage was the only significant finding (95CI 1.09–1.31).

### 3.6. Campaigns

Where immunity gaps exist or there is low routine MCV coverage, the WHO advises immunization campaigns to improve vaccination coverage and close immunity gaps. Six of the fourteen countries had a nationwide intervention (campaign/catch up/follow up for either MMR, MR, or measles) in the three years preceding verification (Table 5). Three countries had no data: Czechia, Lithuania, and Greece.

### 3.7. Generalized Linear Models

The descriptive assessment revealed relevant factors warranting further analysis. Socioeconomic status may be relevant, with 29% (n = 4) of re-established countries in the LMIC grouping. Bordering an endemic country or being classified as a non-island were both statistically significant risk factors. Analysis of tourism ratios indicated that 5 (28%) of re-established countries had a ratio above one, compared to 39 (49%) of NLV countries. As there was no significant finding using the derived tourist pressure ratio, raw annual tourism numbers were included in the model.

In terms of scheduled age of vaccination, first dose is not as variable as second dose. Second dose timing exhibited similar variation among NLV and eliminated countries. Similar levels of vaccination coverage are noted in re-established countries when compared to NLV countries, but risk ratios indicated differences in risk as coverage dropped below 90%, 80%, and 70%.

To assess the contribution of each of these factors to the risk of re-establishment, generalized linear models were used. The reference category was the NLV countries group, and the outcome measured was the risk of re-establishment. The first model indicated a high degree of correlation across vaccination variables, which was to be expected given the degree of relatedness. Models one and two used derived first and second dose coverage categories, <85%, 85–89%, 90–94%, and ≥95%. Controlling for these variables indicated that age at first dose and measles discard rate were associated with lower odds of re-establishment, although this was not statistically significant, which is unsurprising given the small dataset (Figure 9). Countries bordering an endemic country had the greatest risk of re-establishment (OR = 14.6, 95 CI 2.1, 183.2, *p*-value 0.01), whereas countries in the lower-middle income group had an OR of 6.8 but this was not significant (95 CI 0.5, 139.9, *p*-value 0.17). First dose coverage by the categories used was associated with an OR of 3.2, counterintuitively suggesting that an increase in coverage was associated with an increase in the odds of re-establishment. However, higher second dose coverage was associated with a 58% reduction in risk.

As the first dose coverage result was counterintuitive, further investigation of the coverage for the fourteen re-established countries was undertaken. Model 3 derived a simplified vaccination coverage variable with two levels, >90% coverage and ≤90% coverage. This variable was then assessed using a univariate approach. With this approach, higher second dose coverage was protective against re-establishment; however, first dose coverage continued to be associated with an increase in the risk of re-establishment.

To investigate this further, the authors considered the quality of the data included in the models. Seven of the fourteen countries had alternate sources of first dose coverage data available, from MICS (multiple indicator cluster surveys—country-level data collected and collated by UNICEF) surveys (n = 1), District Health Survey (DHS) (n = 2), relevant publications (n = 2), or a PAHO estimate (n = 2) (Table 6).

A second univariate analysis (model 4) was then performed, using, where available, the alternate data sources. Where there was no alternate source available, the WUENIC data were used. This gave very different results (Figure 10). When comparing higher coverage with lower coverage, higher first and second dose coverage were both associated with lower odds of re-establishment. First dose coverage greater than 90% had an OR of 0.38.

The multivariate analysis (model 2) was then performed again using the alternate first dose coverage data. An assessment of collinearity was made between first and second dose coverage, and they were found to be moderately correlated (kappa value = 0.6). Given this correlation, the multivariate analysis was run again, this time excluding the second dose coverage variable. This model indicated that being an upper-middle income country and having a first dose coverage rate >90% were protective, whereas bordering an endemic country and being a lower-middle income country were associated with higher odds of re-establishment. However, this is again hampered by the small dataset; bordering an endemic country provided the only statistically significant *p*-value of 0.008.

These analyses provide an indication of the likely model results where MCV coverage is not overestimated. The results suggest that if the WUENIC estimates are higher than a country’s true coverage rate, the protective effects of both doses are being masked.

## 4. Discussion

The achievement of measles elimination verification by a country is often cause for celebration, and its loss can be devastating. It is thus important that lessons are gleaned from the re-establishment of endemic measles transmission, not only for the country affected, but for global and regional measles elimination efforts. Factors associated with loss of verification status include those that are impossible (non-island status) or very difficult to modify (LMI economic status) in the short term. Other factors, including MCV coverage, are critical for countries to address if they want to bolster their elimination status.

Seven of the eleven factors identified through the narrative review of the literature as possible important factors in the re-establishment of measles transmission were considered important following descriptive analysis. Logistic regression (GLM) provided some predictable and certain unpredictable results. There was a 7% increase in the risk of re-establishment when second dose vaccination coverage fell below 80%, and higher second dose coverage was associated with a 58% reduction in the risk of re-establishment. The socioeconomic status of a country was also associated with risk of re-establishment; countries in the lower-middle income group had a greater risk of re-establishment. Bordering an endemic country was associated with the highest risk of re-establishment.

There is good evidence that lower income countries can achieve elimination success by forming “epi-blocks”, i.e., groupings of countries in the same region with trade, travel, and tourism links forming blocks for coordinated immunization efforts. MECACAR, a coordinated campaign of 18 co-located countries from the Middle East, Caucasus, and the Central Asian Republics, reached 56 million children and most countries involved achieved 95% vaccination coverage in children under 5 years of age [24,25]. Similarly, 21 Pacific Island Countries (PICs) have applied for verification of achieving measles elimination after interrupting a pattern of regular large-scale outbreaks by conducting a multi-country/territory coordinated campaign [19].

Improving the accuracy of MCV coverage estimates is imperative. Maintaining awareness of immunity gaps is necessary to target interventions, such as catch-up campaigns, and to better identify populations vulnerable to increased disease transmission. As demonstrated by this analysis, inflated estimates can mislead. Where there are concerns regarding the timeliness and completeness of source data, household-level surveys such as MICS or DHS may be of value. Vaccine brand, initially planned for inclusion in the analysis, was found to be unreliably available at the country level. These data would be useful for assessing secondary vaccine failure [26].

Equally important to maintaining high vaccination coverage is maintaining a robust surveillance system. These findings are supported by the WHO midterm review of the strategic plan for measles and rubella elimination, which made recommendations to strengthen surveillance, increase vaccination coverage, and increase investment in WHO-led elimination strategies [27]. The Global Measles Rubella Lab Network (GMRLN), the network of 762 laboratories that maintain situational awareness of measles [28], is currently under threat. Costing USD 9 million per year, it was previously funded in full by the US Centres for Disease Control and Prevention and operated by the WHO. The current US political administration has removed funding, leaving the future of the network uncertain [29]. Without the GMRLN, whose laboratories conduct local, regional, and national surveillance, oversight of measles behaviour and transmission will be lost. Prior to the loss of funding, expanding the GMRLN to cover surveillance for other critical pathogens was being considered [28]. Lower income countries will struggle to maintain laboratory capacity locally, making it impossible to meet the surveillance standard needed to confirm elimination, even where elimination has likely been achieved. It is telling that no low-income country has yet had measles elimination verified.

Although it could not be addressed in this study, vaccine hesitancy likely plays a role in declining vaccination rates, particularly following the COVID-19 pandemic. The immediate effect of a re-allocation of immunizer resources resulted in a decrease in rates of childhood vaccination that have not yet rebounded. This may be in part due to higher rates of distrust in vaccination in the post-pandemic era [30].

Limitations are noted in this analysis. This is an ecological study with the attendant weaknesses inherent to ecological associations. The analysis was conducted at country level, naturally losing some of the detail that would be obtained with a finer analysis. WHO data are country-level data from many surveillance systems, and discrepancies in and between the systems are likely. The completeness of the data reported to WHO is unknown. There are limitations to the alternate data utilized; however, a household-level MICS or DHS probably provides a more accurate estimate of true coverage. The data reported by countries on nationwide interventions, such as campaigns, follow up, and catch up, may not be timely, accurate, or complete. It is therefore difficult to make an accurate assessment of immunity gaps or the need for further intervention. The relatively small sample size makes it difficult to draw generalizable conclusions from modelling; therefore, external validity is limited. Whilst some outcomes are logical and likely to be generalizable to a broader context, such as the importance of maintaining high MCV coverage, other factors may be less relevant. There are limitations with assessing the role of tourism, particularly the origins of tourists and whether they are travelling from or through endemic countries. Some population movements, where migrants or migrant workers are crossing borders, are poorly documented, and this probably poses a greater risk. Finally, an assessment of the regression modelling using revised MCV first dose coverage estimates was applied only to the re-established countries. Applying it to the NLV country grouping may reveal variations in the strength of associations.

## 5. Conclusions

Regular global measles resurgences, due to stalled progress towards the six regions’ measles elimination goals, discover country immunity gaps [19]. This analysis of factors associated with the re-establishment of endemic measles transmission emphasizes the importance of high two dose measles coverage and high-quality surveillance as buffers against re-establishment in countries that have been verified to have eliminated measles. This is particularly important for lower-middle income countries, countries bordering endemic countries, and non-island countries, although this analysis shows that even higher income countries are vulnerable if they do not maintain high coverage. It also emphasizes the need for robust data, as current data make an accurate assessment difficult.

The next decade will be a critical time for measles elimination progress and for maintaining elimination status in NLV countries. Regular resurgences are contributing to increased risk and the re-introduction of measles to previously eliminated countries. Without a galvanizing eradication goal, the necessary attendant resourcing, and a coordinated effort by all countries, measles verification in individual countries will remain precarious.

**Key terms** [4]

Eradication: Worldwide interruption of measles transmission in the presence of a robust surveillance system.

Elimination: Local or national interruption of measles transmission for greater than 12 months in the presence of a robust surveillance system.

Endemic measles: Uninterrupted transmission of measles persisting for greater than 12 months in a defined geographical location.

Re-establishment: Detection of a transmission chain that continues uninterrupted for greater than 12 months in a defined geographical region.

Measles discard rate: Rate of suspected cases that have been investigated and discarded as a non-measles using an accredited laboratory test calculated per 100,000 population.

Verified: Interruption of measles transmission in a geographically defined area (country-level for the purposes of this analysis) for a period of 36 months, verified by an independent Regional Commission.

## Figures and Tables

**Figure 1 vaccines-13-01125-f001:**
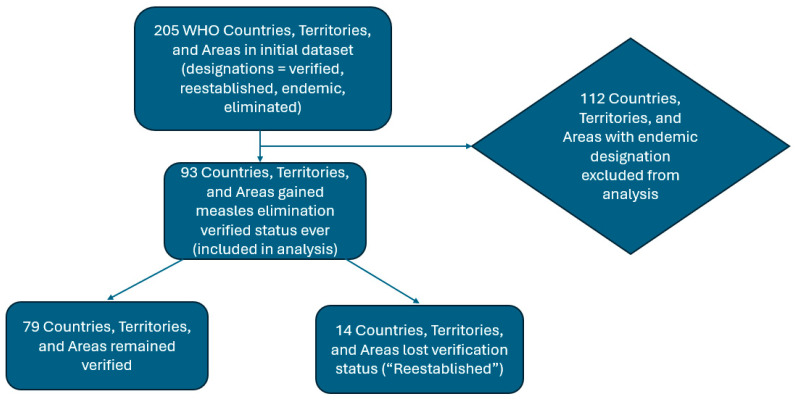
Flow diagram formulating the dataset for analysis. NB. The 11 territories are as follows: (1) occupied Palestinian Territories (oPT)—EMR; (2) American Samoa (US)—WPR; (3) French Polynesia (FR)—WPR; (4) Guam (US)—WPR; (5) Hong Kong SAR (CH)—WPR; (6) Macao SAR (CH)—WPR; (7) New Caledonia (FR)—WPR; (8) Northern Mariana Island (US)—WPR; (9) Pitcairn Island (UK)—WPR; (10) Tokelau (NZ)—WPR; (11) Wallis and Futuna (FR)—WPR.

**Figure 2 vaccines-13-01125-f002:**
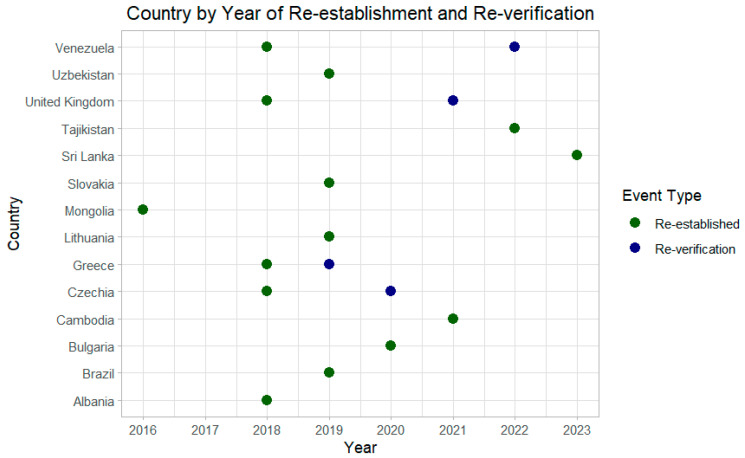
Countries with re-established measles transmission (and re-verification where relevant), by year.

**Figure 3 vaccines-13-01125-f003:**
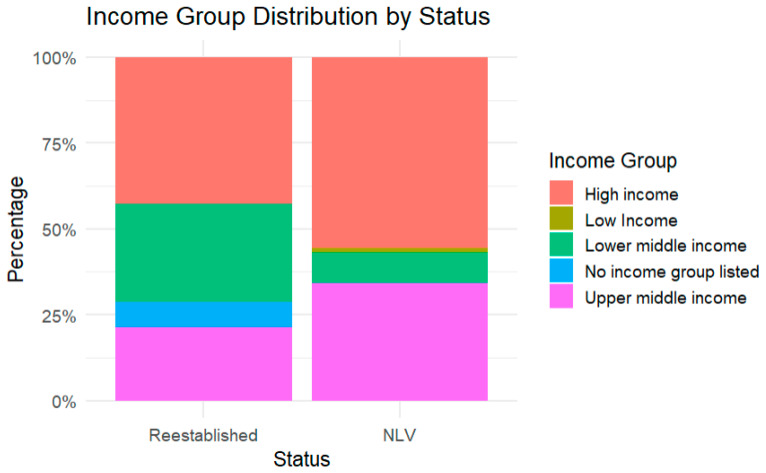
World Bank socioeconomic status for never lost verification (NLV) and re-established countries.

**Figure 4 vaccines-13-01125-f004:**
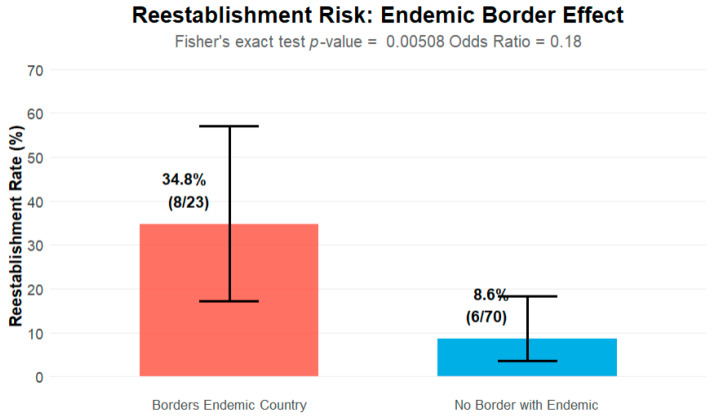
Re-establishment risk and whether countries bordered endemic countries.

**Figure 5 vaccines-13-01125-f005:**
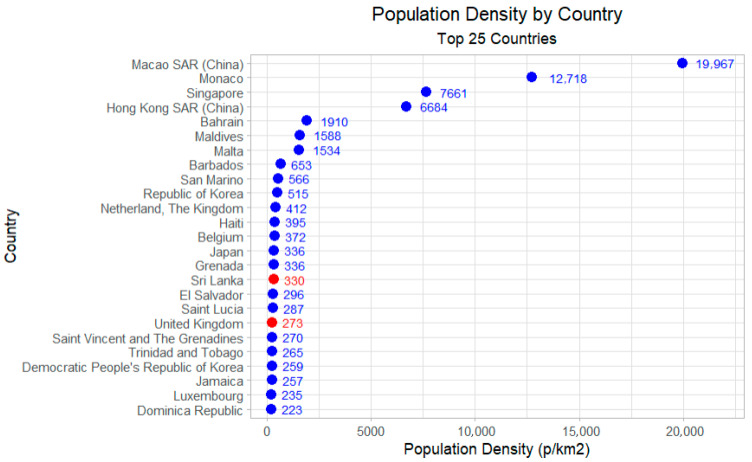
Population density by country: top 25 most populated countries. NB. Figures in blue = NLV countries, figures in red = Reestablished countries.

**Figure 6 vaccines-13-01125-f006:**
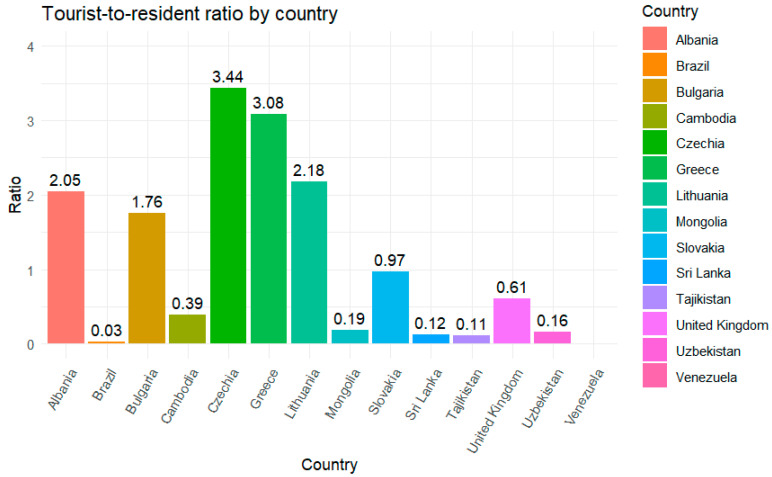
Tourist-to-resident ratio for countries with re-established transmission.

**Figure 7 vaccines-13-01125-f007:**
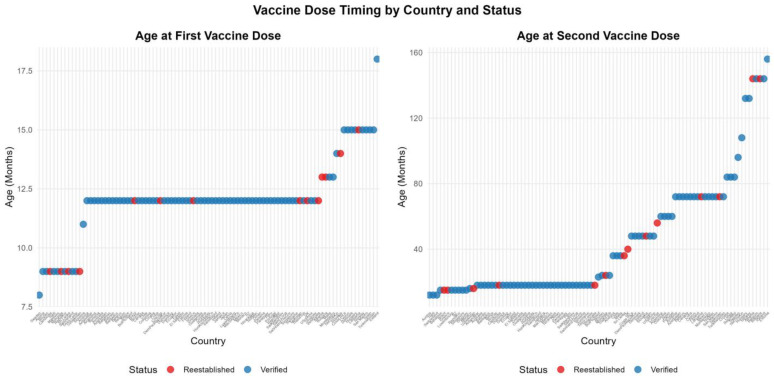
Scheduled age for first and second dose of a measles-containing vaccine for re-established and never lost verification countries.

**Figure 8 vaccines-13-01125-f008:**
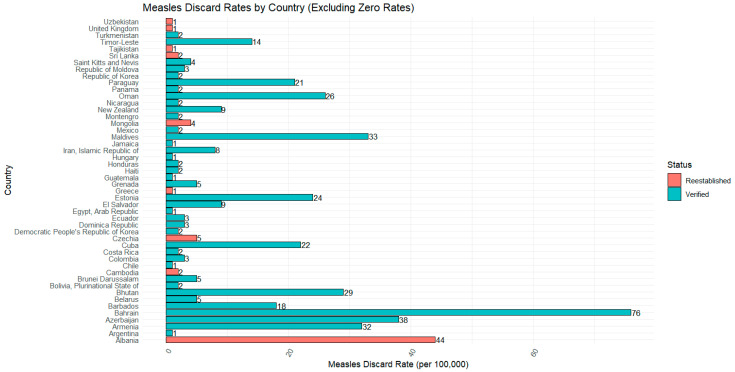
Measles discard rates per 100,000 by NLV and measles re-established countries.

**Figure 9 vaccines-13-01125-f009:**
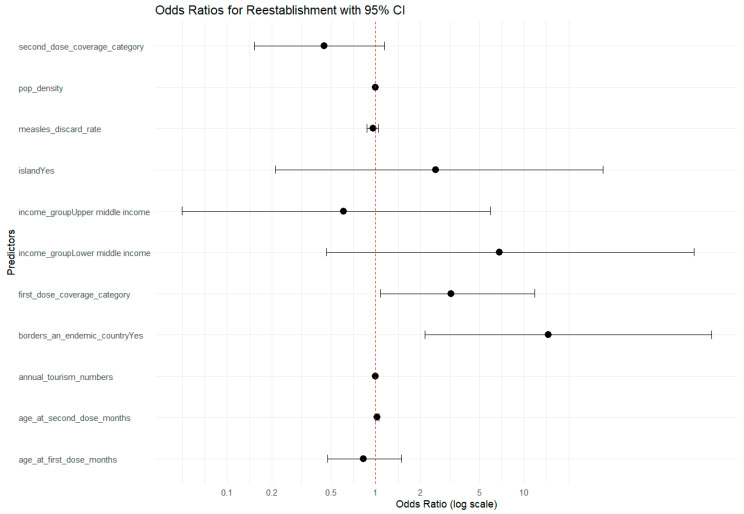
Logistic regression model 2 outcome; odds ratio with 95% confidence intervals.

**Figure 10 vaccines-13-01125-f010:**
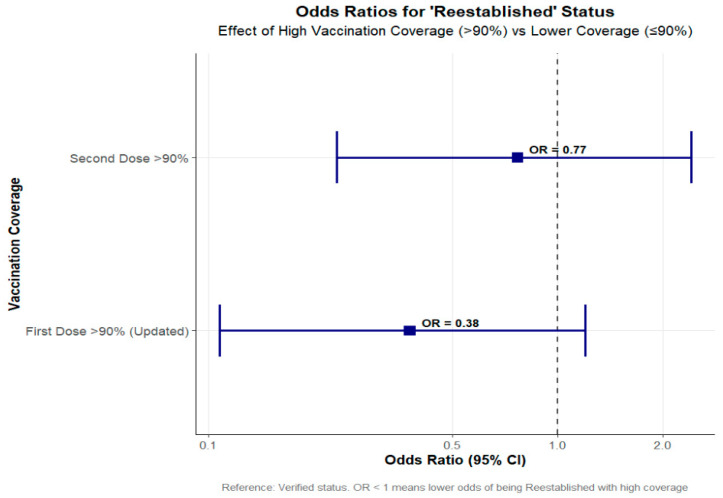
Univariate regression (model 4); first and second dose coverage with revised first dose estimates.

**Table 1 vaccines-13-01125-t001:** Datasets included in analysis.

Dataset	Source	Timepoint	URL (Where Applicable)
Global Measles Elimination Status	WHO	2011–2023	
MCV Campaign Data	WHO	2000–2025	
Measles Reported Cases and Incidence Rates	WHO	2011–2023	
Measles Vaccination Coverage Dose One	WHO	2011–2022	
Measles Vaccination Coverage Dose Two	WHO	2011–2022	
Vaccination Schedule for Measles	WHO	Current Data	Vaccination schedule for measles (https://immunizationdata.who.int/global/wiise-detail-page/vaccination-schedule-for-measles?ISO_3_CODE=&TARGETPOP_GENERAL= (accessed on 29 October 2025))
Tourism Numbers	World Bank	Current Data	https://data.worldbank.org/indicator/ST.INT.ARVL (accessed on 29 October 2025)
Tourism Data/Tourist Tracker	United Nations	2018–2023	https://www.unwto.org/tourism-data/un-tourism-tracker (accessed on 29 October 2025)

**Table 2 vaccines-13-01125-t002:** Factors included in generalized linear models.

Factor	Description
Dose coverage	Country estimates of first and second MCV dose coverage
Population density	The population density value for each country in the dataset
Measles discard rate	The number or suspected measles cases tested and discarded as negative (also a proxy for surveillance quality)
Socioeconomic status	The socioeconomic status for each country in the dataset, per the World Bank designation
Annual tourism numbers	The annual number of tourist visitors to each country, per World Bank data
Age at first and second dose	The age (in months) at which each country schedules first and second dose vaccination of children, considered separately
Island status	Whether a country is classified as an island
Bordering an endemic country	Whether the country bordered at least one country considered endemic for measles

**Table 3 vaccines-13-01125-t003:** Statistical comparison of population density capita/km^2^ for re-established and verified countries.

	Re-Established	Verified
Minimum	2	3
1st quartile	48	28.5
Median	101	777
3rd quartile	108.5	258
Maximum	330	19,967

**Table 4 vaccines-13-01125-t004:** Risk of re-establishment with 10% decreases in MCV coverage for first and second dose.

First Dose Coverage	Risk Ratio (95% Confidence Interval)
<90% coverage	1.11 (0.95, 1.31)
<80% coverage	1.12 (0.95–1.32)
<70% coverage	1.19 (1.09–1.31)
Second Dose Coverage	
<90% coverage	1.00 (0.85, 1.19)
<80% coverage	1.07 (0.91, 1.27)
<70% coverage	0.99 (0.80, 1.23)

**Table 5 vaccines-13-01125-t005:** Nationwide intervention in the three years preceding verification, by country and year.

Country	Verification Year	Intervention Year
Brazil	2016	2014
Bulgaria	2012	2010
Cambodia	2015	2013
Mongolia	2014	2012
United Kingdom	2016	2013
Venezuela	2016	2013, 2014

**Table 6 vaccines-13-01125-t006:** Alternate sources of first dose coverage data, compared with WUENIC estimates.

Country	First Dose Coverage—Alternate	First Dose Coverage—WUENIC	Data Source	Year
Albania	79	95	DHS	2017
Brazil	73	91	PAHO	2021
Cambodia	83	83	DHS	2021
Greece	89.8	97	Publication (Vaccines)	2020
Lithuania	87.2	93	Publication (Medical Sciences)	2023
Mongolia	81.9	98	MICS	2023
Venezuela	68	74	PAHO	2021

## Data Availability

The data presented in this study are included in the article.

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
