# Peer review of "Eliminating Measles: Factors That Contribute to Re-Establishing Transmission"

_vaccines, 2025, doi:10.3390/vaccines13111125_

Round 1
Reviewer 1 Report
Comments and Suggestions for Authors
This manuscript analyzes the risk factors for measles resurgence in countries that lost their elimination status after previously achieving it. Maintaining elimination at the national and regional levels is a critical public health goal, and identifying factors that hinder its sustainability is of great significance. This study has the potential to provide insights that support the global goal of measles eradication. However, the manuscript lacks sufficient description of the data sources and statistical analyses, which raises concerns about reliability. As a result, the comparison between countries that sustained elimination and those with resurgence is not fully developed, and the conclusions show limited novelty. Careful refinement of the data and reconsideration of the manuscript structure are needed.
Major point
Although all data analyzed in this manuscript are publicly available, the description of the resources is insufficient, making it difficult for readers to trace the specific datasets. The statistical methods are also sparsely described, which hinders the assessment of the analyses' validity. Furthermore, for the detailed analysis of vaccination coverage, data were re-extracted from different sources for only seven of the fourteen countries that experienced a resurgence. In contrast, no re-extraction was performed for the 79 countries that maintained elimination. Additionally, the authors did not account for the specific time points (i.e., the year of resurgence) they had defined during the data re-extraction, raising concerns about the appropriateness of the detailed analysis.
Minor point
Introduction
1) The numbers of WHO Member States and measles elimination countries presented in the manuscript differ from those reported by WHO at the end of 2023. Since no references are cited and the source data are not provided, the accuracy of these figures cannot be verified.
(WHO Member States: 194; measles elimination countries: 82)
Minta AA, Ferrari M, Antoni S, et al. Progress Toward Measles Elimination — Worldwide, 2000–2023. MMWR Morb Mortal Wkly Rep 2024;73:1036–1042. DOI: http://dx.doi.org/10.15585/mmwr.mm7345a4
2) The factors assumed by the authors to be related to resurgence are not sufficiently supported by clear evidence or references, and the explanation in relation to the study’s objective is limited.
Materials and Methods
3) The manuscript provides insufficient information on the data sources and links. It would be advisable to provide the raw data as supplementary material.
4) The description of the analytical and statistical methods is not sufficient.
5) Lines 94–95: The vaccine brand was not used in the analysis, which may not be necessary in the manuscript.
Results
6) There is considerable overlap between the content of the figures/tables and the main text, which should be streamlined.
7) Including sample sizes (n) within the figures may help readers understand more easily.
8) The x-axis items in Figures 7, 9, and 10 are not appropriate, and the figures should be reconsidered.
9) The manuscript also lacks an explanation of MICS (Multiple Indicator Cluster Survey).
10) The description of the generalized linear model is insufficient, making it difficult for readers to understand.
11) Lines 130 and 229: Please verify the percentage values.
12) Lines 227–228: The comparison regarding “non-island” is not shown in Figure 3, and the corresponding analytical results do not appear to be provided.
Discussion & Conclusion
13) The factors mentioned by the authors as possible causes of resurgence are important but already well recognized in the context of measles elimination, and thus may not provide new insights. Regarding the decline in vaccination coverage, the discussion could be strengthened by considering potential causes, such as vaccine hesitancy or the impact of the COVID-19 pandemic, etc.
Author Response
Dear Reviewers,
Thank you kindly for your considered review of our manuscript. Please find attached a detailed response.
Warm regards,
Authors

Reviewer 2 Report
Comments and Suggestions for Authors
Gibson et al in this ecological study reported on the important public health topic of reestablishment of measles transmission in coumtries previously considered as having achieved elimination. They authors used international databases (e.g. WHO, Worla Bank, etc) in order to conduct a descriptive, ecological study. They found that vaccination coverage for measles vaccination (2 doses), socioeconomic background and sharing border with an endemic country were risk factors associated with lost of elimination status.
The manuscript is interesting and is dealing with a timely public health topic. It is well- written, but there is space for imorovement.
-Abstact.
The abstract should me concise and free form needless details related to methodology. Please focus on key-results and policy impilations.
-Methods
The authors mentioned that in the context of their ecological study design, they merged data from different sources (WHO, World Bank etc). This point deserves further clarification in the methods section.
-Discussion
Please avoid simple repetition of the results. The limitations of the study should be discussed in more detail. What about the external validity of the results?
Author Response
Dear Reviewers,
Thank you for your considered review. Please find attached a detailed response.
Warm regards,
Authors

Round 2
Reviewer 2 Report
Comments and Suggestions for Authors
The authors have effectively revised the manuscript.